# Novel Variants in the *VCP* Gene Causing Multisystem Proteinopathy 1

**DOI:** 10.3390/genes14030676

**Published:** 2023-03-08

**Authors:** Rod Carlo Agram Columbres, Yue Chin, Sanjana Pratti, Colin Quinn, Luis F. Gonzalez-Cuyar, Michael Weiss, Fabiola Quintero-Rivera, Virginia Kimonis

**Affiliations:** 1Division of Genetics and Genomic Medicine, Department of Pediatrics, University of California, Irvine, CA 92697, USA; 2College of Osteopathic Medicine, William Carey University, Hattiesburg, MS 39401, USA; 3Department of Neurology, University of Pennsylvania, Philadelphia, PA 19104, USA; 4Department of Laboratory Medicine and Pathology, University of Washington, Seattle, WA 98104, USA; 5Department of Neurology, University of Washington, Seattle, WA 98195, USA; 6Department of Pathology and Laboratory Medicine, University of California, Irvine, CA 92697, USA; 7Department of Neurology, University of California, Irvine, CA 92697, USA; 8Department of Pathology, University of California, Irvine, CA 92697, USA

**Keywords:** valosin-containing protein, VCP, IBMPFD, ALS, multisystem proteinopathy-1, MSP1

## Abstract

Valosin-containing protein (*VCP*) gene mutations have been associated with a rare autosomal dominant, adult-onset progressive disease known as multisystem proteinopathy 1 (MSP1), or inclusion body myopathy (IBM), Paget’s disease of bone (PDB), frontotemporal dementia (FTD), (IBMPFD), and amyotrophic lateral sclerosis (ALS). We report the clinical and genetic analysis findings in five patients, three from the same family, with novel *VCP* gene variants: NM_007126.5 *c.1106T>C* (*p.I369T*), *c.478G>A* (*p.A160T*), and *c.760A>T* (*p.I254F*), associated with cardinal MSP1 manifestations including myopathy, PDB, and FTD. Our report adds to the spectrum of heterozygous pathogenic variants found in the *VCP* gene and the high degree of clinical heterogeneity. This case series prompts increased awareness and early consideration of MSP1 in the differential diagnosis of myopathies and/or PDB, dementia, or ALS to improve the diagnosis and early management of clinical symptoms.

## 1. Introduction

Multisystem proteinopathy 1, or inclusion body myopathy associated with Paget’s disease of bone and frontotemporal dementia, is a rare autosomal dominant disease caused by mutations in the *VCP* gene [1,2]. It manifests clinically as myopathy and/or Paget’s associated bone pain that starts in the mid-thirties and dementia in the late fifties [3,4]. Proximal muscle weakness progresses to involve limbs and respiratory muscles [5]. PDB causes increased bone turnover in focal areas, leading to bone pain and pathologic fractures [6]. FTD manifests as apathy, compulsive behaviors, disinhibition, auditory comprehension deficits, word-finding difficulties, and hyperorality [5]. No treatment is available for MSP1; however, active research efforts are ongoing.

### 1.1. Pathology of MSP1

Inclusion body myopathy is characterized by progressive adult-onset muscle weakness. Affected individuals first experience difficulty in walking and performing overhead activities [2]. Although 90% of affected individuals have proximal limb-girdle weakness, other muscle groups have been found to become involved over time progressing to respiratory failure, cardiomyopathy, and dysphagia [5,7,8,9,10,11,12,13]. The serum creatine kinase (CK) level is usually normal to mildly elevated (mean: 195 IU/L; range: 40–1145 IU/L; normal range: 20–222 IU/L). Electromyography (EMG) shows changes consistent with a myopathy, as well as superimposed active and chronic denervation [2]. Histologically, characteristic findings in skeletal muscle include rimmed vacuoles/sarcoplasmic aggregates containing the same proteins that aggregate in the brains of patients with neurodegenerative disease, such as tau, amyloid, and TDP-43 [5]. Deposits can also be highlighted with other proteins, including ubiquitin and p62, via immunohistochemical staining [14]. Other myopathies present similarly to IBM and have often led to misdiagnoses, including limb-girdle muscular dystrophy, GNE-related myopathy, sporadic inclusion body myositis (sIBM), facioscapulohumeral muscular dystrophy, ALS, scapuloperoneal myopathy, myotonic dystrophy, myofibrillar myopathies, skeletal muscle channelopathies, and spinobulbar muscular dystrophy [2,5,15].

Paget’s disease of bone is a localized disorder of bone remodeling caused by an imbalanced activity of osteoclasts and osteoblasts, resulting in the formation of poor-quality bone prone to pathologic fracture. Affected individuals present with spine or hip pain, body tenderness, long-bone or cranial-bone deformity, reduced height, pathologic fractures, or hearing loss. Rare complications include kidney stones, osteosarcoma, and high-output heart failure due to the formation of arteriovenous shunts in bone [5,16,17,18]. Elevated serum alkaline phosphatase (ALP) (mean: 359 IU/L; normal range: 30–130 IU/L) is usually observed. Urine concentrations of pyridinoline (PYD) (mean: 153 IU/L; normal: 31.3 IU/L) and deoxypyridinoline (DPD) (mean: 40 IU/L; normal: 6.8 IU/L) are elevated in some case; however, they are not required for diagnosis [2]. Screening radionuclide scans show focally increased bony uptake in the skull, pelvis, and spine [2,17,19]. Skeletal radiographs reveal coarse trabeculation, cortical thickening, and spotty sclerosis in the skull, scapula, pelvis, and spine [5].

Frontotemporal dementia encompasses changes in behavior, language, executive control, and motor symptoms accompanied by focal degeneration of the frontal and/or temporal lobes. The pathological entities of FTD are classified according to the observable histopathological cytoplasmic or nuclear protein inclusions. The average onset is 55 years; however, some cases younger than 30 years have been documented [5]. Early behavioral changes include disinhibition, apathy, loss of empathy, hyperorality and compulsive behaviors, and generally lack cranial nerve, sensory, cerebellar, pyramidal, and extrapyramidal motor findings [20]. In later stages, disinhibition and compulsive behaviors regress, while apathy worsens indicating the progressive degeneration of the medial frontal cortex. As the disease progresses, focal atrophy of the frontal or temporal lobes can be appreciated in 50–65 percent of patients [21]. The presence of abnormal social conduct, stereotyped behaviors, visual and special defects, memory issues, and apathy in neuropsychological testing can aid in an FTD diagnosis. CT or MRI brain scans are helpful in identifying structural pathology and cortical atrophy. PET imaging has shown benefits in differentiating FTD from Alzheimer’s disease and was found to be more sensitive than MRI in identifying FTD in early stages [3]. However, its role as a diagnostic tool in MSP1-associated FTD prompts further study.

Amyotrophic lateral sclerosis (ALS) is a neuronal degeneration disorder of the corticospinal tract affecting both upper and lower motor neurons (UMNs and LMNs). Motor neuron degeneration and death, with gliosis replacing the lost neurons, are characteristics of ALS. Histologically, intracellular inclusions, Bunina bodies, and TDP-43 accumulation can be seen in the degenerating neurons and glia. Inclusions that stain positively for ubiquitin are hallmarks of ALS. The most common presentation of ALS is painless asymmetric limb weakness, slow movement, incoordination, stiffness, spastic gait, and poor balance. Weakness of the diaphragm can also be seen as progressive dyspnea culminating in dyspnea at rest and talking. Diagnosing ALS involves addressing progressive UMN or LMN signs and symptoms in one limb or body segments or progressive LMN signs and symptoms in at least two body segments and the absence of electrophysiologic, neuroimaging, and pathologic evidence of other disease processes [22]. The presence of acute and chronic denervation and reinnervation by electromyography in multiple body segments is also vital in diagnosing ALS. Although muscle biopsy is not a routine part of diagnosing ALS, it is performed when myopathy is suspected. The findings in ALS are nonspecific, demonstrating evidence for a combination of active and chronic denervation, including groups of reinnervated as well as angular esterase-positive active denervated muscle fibers.

### 1.2. VCP Variants

The *VCP* gene encodes for the protein p97, a type II ATPase associated with various cellular activities (AAA+) that catalyze ATP hydrolysis to perform its multiple cellular functions. p97, with its various cofactors, is involved in autophagy, cell cycle regulation, Golgi and nuclear membrane reassembly, and ER-associated degradation (ERAD) [23,24,25,26,27,28]. p97 comprise approximately 1% of the total cytosolic proteins [29].

Structurally, p97 is a homo-hexamer with each monomer composed of an N-terminal domain (NTD), two ATPase domains (D1 and D2), and a C-terminal domain [30]. The NTD and C-terminal domains regulate ATPase function, while the D1 and D2 domains mediate ATP hydrolysis [31,32,33,34,35,36,37,38,39]. Most MSP1-associated p97 variants are clustered on the NTD or the N-D1 linker, which enhances the p97 ATPase activity and dysregulates cofactor binding [38,40,41]. Currently, there are over 65 heterozygous missense variants known to cause MSP1. The most common *VCP*/p97 mutations include R155H, R155C, R159H, and R93C [4]. To date, additional *VCP* variants have been reported, a few of which were classified as variants of unknown significance that were not previously associated with human disease [3,42,43,44,45,46,47,48,49,50,51]. Here, we report multiple previously undescribed likely pathogenic variants in the *VCP* gene that were identified in five patients, three from the same family.

## 2. Case Report

Clinical features from each proband are summarized in Table 1. All participants signed an informed consent, including the release of medical records, which consisted of genetic tests, diagnosis, laboratory studies, imaging, electromyograms, nerve conduction studies, and biopsies.

### 2.1. Family 1 Proband 1 (c.1106T>C; p.Ile369Thr)

The proband (III:3) (Figure 1) was a 71 year old male of South Asian Indian ancestry, who presented with generalized muscle weakness, such as difficulty getting up from the floor, climbing stairs, and lifting objects at age 60 years. At age 62 years, he noted proximal arm weakness. At age 67 years, he had frequent falls due to the fact of his knees “giving out”, decreased overhead activity in the right arm, and difficulty opening bottle caps. He denied pain in his back or hips. On physical examination, he had bilateral scapular winging, with the right side more prominent (Figure 2A). He had to use both hands and knees from a squatting position to standing; however, he had a normal gait. Strength testing revealed proximal shoulder weakness (Medical Research Council (MRC) grade 3) and moderate distal leg weakness (ankle dorsiflexion, MRC 4). Tendon reflexes were 2+/4. His upper and lower extremities’ weakness was more significant proximally. Mild symmetric facial motor weakness was also noted. His forced vital capacity (FVC) of 3.3 L and best maximal inspiratory pressure of −35 cm H_2_O were suggestive of diaphragmatic insufficiency. The proband demonstrated normal mental status and fluent speech.

Laboratory results revealed a CK of 117 IU/L, and the cytosolic 5′-nucleotidase 1A (NT5c1A) antibody testing was negative. The EMG revealed spontaneous activity in the form of fibrillation potentials, a mix of large and small amplitude motor units, reduced recruitment in the right tibialis anterior and medial gastrocnemius, and early recruitment in the right vastus lateralis. Muscle biopsy of the right thigh revealed muscle fiber size variability (Figure 2B), with patchy areas of marked atrophy. Degenerating and regenerating fibers, foci of myophagocytosis, angulated fibers, and numerous internally placed nuclei were observed. There were rare vacuoles. No signs of mitochondrial dysfunction (e.g., ragged, red fibers) were seen. There was no evidence of endomysial inflammation.

The MRI findings of bilateral thighs showed atrophy and fat replacement in the anterior and posterior compartments (Figure 2C).

The proband’s medical history was significant for a small disc herniation at L4-5, thalassemia trait, and a longstanding right lower extremity radiculopathy. His father (II:3) suffered from heart disease in his mid-70s, whereas his mother (II:4) had unexplained distal lower extremity weakness at the age of 75 years. The maternal uncle (II:6) and aunt (II:7) both suffered from dementia in their 50s and mid-70s, respectively.

Genetic testing was conducted in proband III:3 only. An initial neuromuscular panel of 80 genes was unrevealing. The clinical exome sequencing revealed a novel c.1106T>C *VCP* variant resulting in a missense change in exon 10, which led to a protein substitution of isoleucine with threonine at position 369 (p.I369T) (Table 2).

### 2.2. Family 2 (c.478 G>A; p.A160T)

The proband (IV:12) (Figure 3) was a 57 year old male, who was incidentally diagnosed with Paget’s disease of bone due to the presence of an elevated ALP and bone lesions seen on an X-ray when he was in his mid-40s. At age 50 years, he reported lower extremity weakness, difficulty standing up from a sitting position, trouble climbing stairs without using a banister, chronic back pain, and fatigue. Upon physical exam, the upper extremity strength revealed no weakness, scapular winging, or atrophy bilaterally. The MRC scale showed a 5/5 bilateral elbow, wrist and shoulder extension and flexion, with normal grip strength. The lower extremity strength revealed an ability to perform a heel-to-toe walk and an MRC scale of 4+/5 in left hip flexion, extension, and abduction; 3−/5 left knee extension; and 4/5 right knee extension. Other physical exam findings includes substitute motions, such as leaning forward or rocking while performing sit-to-stand tests, suggestive of mild proximal muscle weakness. He was able to repeat and name objects and follow complex commands without any difficulty. He had intact memory for recent and remote events and had fluent speech. The electromyogram revealed a mild nonirritative myopathy without abnormal spontaneous activity involving the left bicep muscle only, despite testing in apparently weak muscles. A transthoracic echocardiogram revealed mild focal septal hypertrophy and mild late systolic prolapse of the posterior leaflet. No muscle biopsy was performed. The laboratory results revealed ALP levels at 146 IU/L and CK at 318 IU/L.

Plain radiographs showed thickened trabeculation with increased sclerosis of the T11 and T12 vertebral bodies consistent with Paget changes. There was also a mild degenerative disc disease at the thoracolumbar junction and fusion of the sacroiliac joint inferiorly. The CT showed an abnormal bone matrix of T11 and T12 vertebral bodies and posterior elements, highly suspicious for Paget’s disease. Ultrasound of the thyroid showed a left solitary nodule highly suggestive of chronic thyroiditis. MRI scans showed mild edema of the right vastus lateralis and medialis muscles, bilateral biceps femoris, and bilateral posterior compartment musculature. In addition, generalized fatty muscle atrophy was observed with advanced atrophy on the left vastus medialis, moderate atrophy involving the right posterior compartment and bilateral gluteus minimus muscles, and mild atrophy on the bilateral tensor fascia lata.

Given the combination of PDB and myopathy, a comprehensive neuromuscular disease NGS gene panel and deletion/duplication analysis were performed, which revealed a novel variant *c.478 G>A* that resulted in the protein substitution of alanine with threonine at position 160 (*p.A160T)* in the *VCP* gene. Subsequently, whole exome sequencing was conducted, which did not reveal any additional pathogenic variants.

The proband’s mother (III:3) was diagnosed with IBM in her 50s, with rimmed vacuoles noted in her muscle biopsy. She also had delayed responses in conversations and confusion about her personal location later in her life, suggestive of dementia. She died at 72 years of age from recurrent aspiration and pneumonia. The proband’s father (III:2) had metastatic thyroid cancer. The proband has a 54 year old sister (IV:2) with unilateral retinoblastoma, thyroid cancer, anxiety, depression, and progressive memory loss at 52 years and was later identified with the familial *VCP* variant. She has been diagnosed with FTD and muscle weakness, and also has bone pain. No imaging or further testing was reported. A 48 year old sister (IV:7) with muscle weakness and Hashimoto’s disease also tested positive for the familial variant, and a 46 year old sister (IV:9) was determined to be a noncarrier after genetic testing. The other siblings (IV:1 and IV:4) have not been tested for this mutation. Mixed European and Ashkenazi Jewish heritage is reported on both sides of the family.

### 2.3. Family 3 Proband 3 (c.760A>T; p.Ile254Phe)

The proband (III:3) (Figure 4) was a 49 year old male of Chinese ancestry that initially presented with left proximal muscle weakness that progressively worsened. He also reported dyspnea on exertion while climbing stairs at age 45 years. He has a past medical history of Paget’s disease, diagnosed at age 45 years. He noted easy fatigue with exertion with climbing stairs that was not associated with shortness of breath, chest pain, muscle pain, muscle atrophy, or focal weakness. His gait and muscle tone were within normal limits upon the physical exam. His muscle strength was reduced in the left biceps 4+/5; left flexor digitorum profundi (FDP) I/II 0/5; right FDP I/II 4+/5; left FDP III/IV 3/5; right FDP III/IV 4+/5; left hip flexor 4/5; right hip flexor 3+/5; and left ankle dorsiflexors 4+/5. He had a full range of motion in his hips and shoulders bilaterally. He also had mild scapular winging bilaterally.

The laboratory results indicated a significantly elevated CPK, ranging from 810 to 2378 mg/dl, and normal acetylcholine receptor (AchR) binding antibodies, muscle specific kinase (MuSK), testosterone, and antinuclear antibody (ANA) panel. In addition, his ALP level was at 263 IU/L, AST at 59 IU/L, ALT at 107 IU/L, and aldolase at 11.5 IU/L. The EMG studies showed evidence of an irritative myopathy, specifically involving the left vastus lateralis and triceps brachii. His motor nerve conduction studies and sensory nerve conduction studies were within normal limits. Biopsy of the left thigh muscle revealed mild myofiber size variation, focal myofiber atrophy, and very rare myofibers containing rimmed vacuoles (Figure 5B) or subsarcolemmal TAR DNA-binding protein 43 (TDP43)/phospho-TDP43 immunoreactive aggregates. Rare scattered myonecrotic and regenerating fibers were also noted. There was no definite evidence of endomysial fibrosis and fatty replacement, neither was there diagnostic evidence of vasculitis, neurogenic atrophy, necrotizing, or inflammatory myopathy (Figure 5A–D).

Pelvic radiograph showed an increased density of the left iliac bone and acetabulum consistent with Paget’s disease. Bilateral hip joint space narrowing was also present and consistent with chondromalacia. An asymmetric narrow left sacroiliac joint with sclerosis on the iliac side was also noted, including mild facet degenerative joint disease at L4-5 and L5-S1. Nuclear bone scan of the whole body indicated abnormal focal uptake at the left mandible, left hemipelvis, multiple levels on the thoracic and lumbar spine, and the proximal left third of fibula reflecting Paget’s disease. An MRI demonstrated muscle atrophy and fatty infiltration of the left vastus lateralis, vastus intermedius, distal vastus medialis, and short head of biceps femoris muscles with severe muscular atrophy and fatty replacement. Mild edema of the rectus femoris muscle with muscle atrophy was noted from the abovementioned muscles. In addition, there was mild edema of the proximal medial gastrocnemius and distal semimembranosus muscles.

A comprehensive neuromuscular disorder panel sequence analysis and deletion/duplication testing revealed a novel *c.41A>G* variant resulting in a protein substitution of lysine with arginine at position 14 *(p.Lys14Arg*).

The proband’s paternal grandfather (I:1) had a history of muscular dystrophy and Alzheimer’s disease prior to his demise at age 68 years. His father (II:2) suffered from a presumed heart disease leading to his early death at 47 years. His 78 year old mother (II:3) and two sisters have not reported musculoskeletal, bone, or cognitive problems. He has a healthy 18 year old son (IV:4).

## 3. Discussion

Muscle weakness was present in all four individuals in this report, though the patterns varied among the individuals, including bilateral proximal UE or LE muscle weakness, unilateral proximal UE weakness, and diaphragmatic insufficiency. Of the two patients who underwent muscle biopsies in this study, one demonstrated characteristic findings, including vacuoles and subsarcolemmal TDP-43 aggregates, whereas the other showed marked atrophy without rimmed vacuoles in the quadricep muscle. Only the muscle biopsy from proband 1 showed evidence for concomitant lower motor neuron disease based on the presence of angulated muscle fibers. As in previous studies, we noted the full spectrum of CK values. One patient had normal CK at 117 IU/L, one had a slight elevation at 318 IU/L, and one had a markedly elevated level up to 2378 IU/L. The EMG findings in our cohort showed variable focal myopathic findings with the proband from family 2 showing variable findings in the biceps femoris, proband 1 in the vastus lateralis, and proband 3 in both the biceps femoris and vastus lateralis muscles. The lack of EMG changes in other muscles could be due to the fact of a sampling error in regard to the muscle selection, as well as the patchy nature of myopathy.

On imaging, an MRI of the thigh muscles of the proband from family 2 and proband 3 revealed mild edema in the musculature, including generalized muscle atrophy and fat infiltration. The proband from family 2 and proband 3 had an onset of PDB at age 44 and 45 years, respectively. Both probands from family 2 and proband 3 had elevated ALP values; however, they had denied bone pain, swelling fractures, or hearing loss. The plain radiograph revealed thickened trabeculation and sclerosis of the vertebral bodies in the proband from family 2 and increased density of the pelvic bones in proband 3. Other characteristic findings included abnormal bone matrix in the CT scan and focally increased bony uptake in the radionucleotide scan.

None of the individuals in this study who were below the mean age of FTD onset (55.9 years) showed symptoms. However, two of the probands had a significant family history of dementia (Table 1).

Three affected members from family 2 had thyroid disease. The proband had chronic thyroiditis, sister IV:7 had Hashimoto’s disease, while sister IV:2 and their father (III:2) had thyroid cancer. A higher rate of rare tumors was associated with VCP variants, some of which developed before the onset of MSP1 symptoms [52]. The c.478G>A variant seen in family 2 might have potentially caused an increased risk for thyroid disease; however, future investigations on this association are warranted.

Most *VCP* mutations are located at the interface between the N and D1 domains in the tertiary structure of the wild type p97 [5,23,42,53]. A total of 11 out of 17 exons have been identified with *VCP* mutations; the only exons that have not been reported with a mutation thus far are exons 1, 8, 9, 13, and 15 [5]. Studies have shown that the *VCP*/p97 mutations at different loci will lead to variations in functional outcomes at the molecular and cellular levels and, consequently, different phenotypes [5,8,42,54]. A mutation at the arginine residue at codon 155 accounts for three-quarters of reported cases. Among mutations affecting amino acid R155, R155C was previously thought to be associated with an earlier onset of symptoms and a reduced mean survival when compared to R155H [8]. A study with a larger dataset confirmed these findings [42]. The R159C variant, on the other hand, is associated with a later onset of myopathy compared to R155H, R155C, R155P, and L198W and has a protective effect against PDB [5,55,56,57]. Recently, the R159H variant found in five families with Hispanic ancestry was also associated with a later onset of manifestations and the rare occurrence of PDB seen in one patient. FTD was the most prevalent feature reported, particularly frequent in females, reported in 72% of individuals [58].

A study has demonstrated elevated ATPase activity in ten *VCP* variants compared to the wild type in which the A232E and R155C variants had the highest activities [8,34]. The elevation in A232E was higher and, interestingly, correlated with more severe phenotypes, such as fractures and PDB, at an earlier age and a more aggressive form of myopathy [28,34,59]. Unfortunately, other clear genotype–phenotype correlations appear to be limited. A recent study has suggested that due to the late-onset nature of MSP1, environmental factors and host–microbiome interactions can contribute to the variability of the disease. Genetic modifiers, such as APOE4 alleles and ATNX2 repeats, have also been found to link to the neurodegenerative phenotype of the disease [42,60].

In our study, family 2 has a c.478G>A variant that resulted in a A160T in exon 5, which correlates to the p97 N-terminal domain and is the most frequently mutated exon identified thus far. Indeed, a change from a hydrophobic amino acid to a hydrophilic amino acid in the NTD can potentially affect the binding of cofactors within this domain and, subsequently, affect downstream ATP hydrolysis. The p.A160 amino acid is highly conserved among most species (Table 2), and substitutions in the proximity (e.g., p.R155H, p.G157G, and p.R159H) are reported to be pathogenic [43,61,62]. While the classification in ClinVar is conflicting with four entries, three were classified as VUS, and one as pathogenic, and the PolyPhen prediction was benign. There are other criteria that supports this variant’s pathogenicity, including segregation in multiple affected members in this family, absence in the gnomAD V.2.1.1 database, and deleterious prediction by MutationTaster (v2021) software. The MutationTaster algorithm uses various information sources, such as evolutionary conservation, functional annotations, and experimental data, to make its predictions. The tool assigns a score to each variation, indicating the probability of it being deleterious, benign, or unknown. The final prediction is made based on a combination of these scores (Table 2).

Proband 3 has a c.760A>T variant that resulted in a substitution of isoleucine to phenylalanine in codon 254 in the D1 domain and in exon 7. This amino acid is highly conserved among all species (Table 2). Mutation in this domain is potentially more detrimental, because the D1 domain’s catalytic effect is critical for the formation of hexamers [28,63]. Variant p.A232E, as mentioned above, is in the proximity of p.I254F and is associated with more severe phenotypes. Other nearby variants, p.T262A and p.T262S, are associated with the FTD phenotypes. An uncommon presentation of primary progressive aphasia was also present in one individual with the p.T262A variant [2,55,64]. In ClinVar, this variant is classified as VUS in an individual with IBMPFD and one as pathogenic; however, there are other criteria that support this variant as likely pathogenic, including its absence in the gnomAD V.2.1.1 database and the PolyPhen and MutationTaster (v2021) predictions as deleterious and possibly damaging, respectively.

Proband 1 has a c.1106T>C nucleotide change leading to a substitution of hydrophobic isoleucine with hydrophilic threonine in codon 369 in exon 10. Probands 1 and 5’s variant lies in the heat-enhanced ATPase D1 domain. This amino acid is highly conserved among all species (Table 2). The missense variant in this region is highly intolerant to missense variation (high constraint region in the DECIPHER database) in the D1 oligomerization domain [65]. Although I369 lies in the D1 ATPase site, there are several well studied variants nearby, suggesting that the N-terminal domain was more affected than the catalytic site of the D1 domain [66,67]. In addition, the 3D structure of p97 shows significant interaction between R155 and N387. However, mutation in the N387 can alter the N-terminal conformation, potentially affecting binding interactions within the N-terminal domain [66]. In ClinVar, this variant is classified as VUS in three individuals with IBMPFD or Charcot–Marie–Tooth disease type 2; however, there are other criteria that support this variant’s likely pathogenicity, including its absence in the gnomAD V.2.1.1 database and the PolyPhen and MutationTaster (v2021) predictions were deleterious and possibly damaging, respectively. Furthermore, the variant has previously been observed to segregate in two affected siblings (Leiden Open Variation Database (LOVD), Individual ID 00376137, Leiden, The Netherlands; https://github.com/LOVDnl/api.lovd.nl accessed on 6 February 2023)

All of these variants were originally reported as a variant of unknown significance (VUS), because they were not previously published in association with MSP1 and, thus, can be limiting. However, each variant was consistently predicted to be damaging by multiple in silico tools, and it is highly conserved with a moderate amino acid change, absent in normal controls, and segregates with the phenotype. According to the standards and guidelines for interpreting sequence variants from the joint consensus recommendation of the American College of Medical Genetics and Genomics (ACMG) and the Association for Molecular Pathology (AMP) [68], these variants fall within the following categories: (a) PM1—located in a mutational hotspot and/or critical and well-established functional domain (e.g., active site of an enzyme) without benign variation; (b) PM2—absent from controls; (c) PP1—cosegregation with disease in multiple affected family members in a gene definitively known to cause the disease (Family 2); (d) PP3—multiple lines of computational evidence support a deleterious effect; and (e) PP4—patient’s phenotype or family history is highly specific for a disease with a single gene etiology. Thus, there is enough evidence of pathogenicity according to the ACMG guidelines: likely pathogenic—2 moderate (PM1–PM6) and ≥2 supporting (PP1–PP5). Future functional in vitro assays may help provide further evidence of their pathogenicity [42].

There is currently no definitive treatment available for MSP1. The mainstay management is supportive care, which includes physical therapy, occupational therapy, speech-language pathology, and respiratory therapy. MSP1 patients with PDB can be treated with bisphosphonates to help relieve the pain, deformity, and complications resulting from progression [69]. It is essential for these patients to receive referrals to endocrinology or rheumatology for proper PDB management. Since MSP1 affects multiple body structures and functions, monitoring and aiding patients with their activities of daily living and quality of life should be a priority. Mental health is an important aspect that needs to be addressed in MSP1 patients, as they have a higher risk for stress, depression, and anxiety due to the nature of their disease and its progression. Appropriate therapeutic support has been shown to improve patients’ quality of life [3]. MSP1 patients could also experience heart failure and cardiomyopathy; thus, periodic surveillance and referral to cardiology are recommended [3].

The *VCP* gene represents a wide array of variants causing MSP1. Currently, genetic testing remains the definitive diagnostic tool for MSP1. It is therefore recommended to test the *VCP* gene in patients with one or more MSP1 features. Due to the autosomal dominant inheritance pattern of this disease, at-risk family members are encouraged to undergo genetic counseling and testing. Asymptomatic relatives and those planning to start a family are recommended for genetic counseling to assess the risks for themselves and/or their offspring. All five individuals presented here were alive at the time of writing the manuscript; however, their symptoms have been progressing.

## Figures and Tables

**Figure 1 genes-14-00676-f001:**
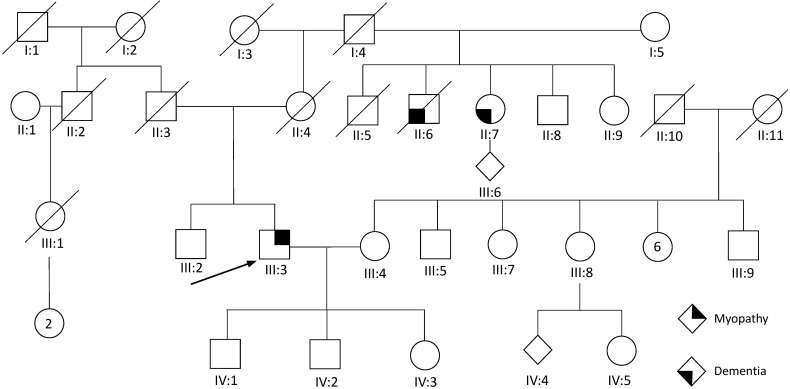
Pedigree of proband 1. Proband (III:3) is indicated by the arrow.

**Figure 2 genes-14-00676-f002:**
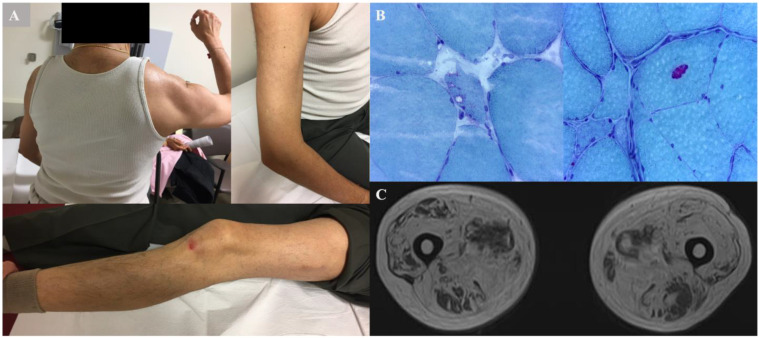
Clinical signs, biopsy, and MRI findings from proband 1 (III:3): (**A**) bilateral scapular winging prominent on the right, with humeral muscle atrophy and bilateral thigh muscle atrophy; (**B**) thigh biopsy stained with modified Gomori Trichrome on a cryosection, where variability in the muscle fiber size with areas of fiber size variation and vacuoles can be seen; (**C**) MRI of bilateral thighs demonstrating prominent atrophy and fatty replacement in both the anterior and posterior compartments.

**Figure 3 genes-14-00676-f003:**
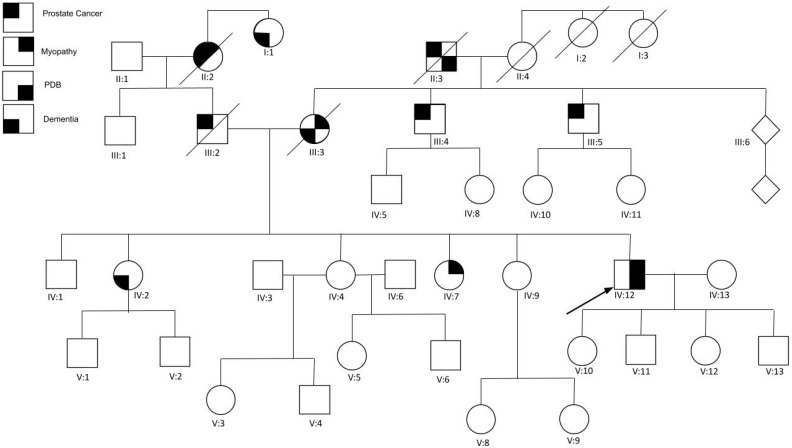
Pedigree of family 2. The proband (IV:12) is indicated by the arrow.

**Figure 4 genes-14-00676-f004:**
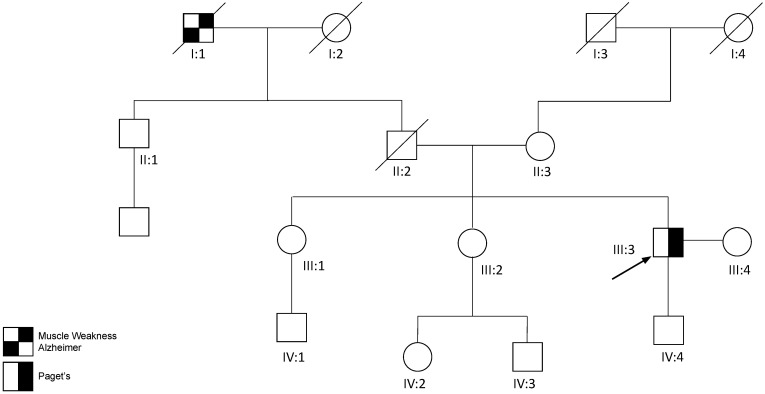
Pedigree of family 3. The proband (III:3) is indicated by the arrow.

**Figure 5 genes-14-00676-f005:**
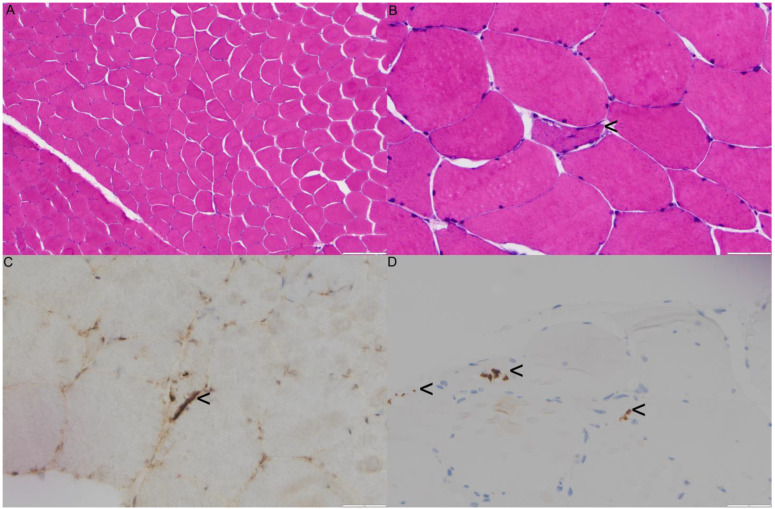
Biopsy findings for proband 3 (III:3). (**A**) Representative low-power magnification of an H&E-stained cryosection of skeletal muscle demonstrating fascicular architecture with mild myofiber size variation without definite endomysial fibrosis, fatty replacement, perimysial pathology, vasculitis, inflammatory or necrotizing myopathy, or perifascicular atrophy/necrosis. Rare areas with marked myofiber atrophy and very rare degenerating myofibers were also noted (not shown). (**B**) High power magnification of the same H&E-stained section as seen in (**A**) showing an atrophic fiber with mildly basophilic sarcoplasm with subsarcolemmal rimmed clefts/defects with granular eosinophilic material (rimmed vacuoles) (black arrowhead). (**C**) High-power magnification of a TDP43 immunohistochemical (IHC) stain performed on a cryosection demonstrating subsarcolemmal aggregates (black arrowhead). (**D**) Phospho-TDP43 (Ser409/Ser410) (1D3) IHC stain on formalin-fixed paraffin embedded (FFPE) tissue showing staining compatible with sarcoplasmic aggregates and rimmed vacuoles (black arrowheads).

**Table 1 genes-14-00676-t001:** Clinical features of the MSP1 patients with novel *VCP* gene variants.

Case	Age (Y)	Sex	DNA Mutation	ProteinMutation	Onset, Myopathy (Y)	Onset, PDB(Y)	Onset, FTD (Y)	ALP (Normal 30 to 130 IU/L)	CK (Normal 20 to 222 IU/L)	Family History of Associated Disease/Genetic Confirmation Not Performed
Family 1 Proband 1 (III:3)	71	M	c.1106T>C	I369T	60	-	-	71	117	Dementia (maternal uncle (II:6) and aunt (II:7))
Family 2										
Proband 2(IV:12)	56	M	c.478G>A	A160T	50	44	-	146	318	Inclusion body myopathy and dementia (mother III:3)
(IV:2)	54	F			-	-	52	-	-	
(IV:7)	48	F			45	-	-	44	83	
Family 3 Proband 3 (III:3)	49	M	c.760A>T	I254F	49	45	-	263	2378	Muscular dystrophy and Alzheimer’s disease (paternal grandfather I:1)

[-] No reported data (data not available).

**Table 2 genes-14-00676-t002:** Prediction of the likely impact of novel variants.

	Proband 1	Family 2	Proband 3
**Transcript ID**	NM_007126.5(VCP):c.1106T>C	NM_007126.5(VCP):c.478G>C	NM_007126.5(VCP):c.760A>T
**Protein**	NP_009057.1:p.Ile369Thr	NP_009057.1:p.Ala160Pro	NP_009057.1:p.Ile254Phe
**Protein domain of variant localization**	D1	NTD	D1
**Type of single nucleotide variant**	Missense	Missense	Missense
**Allelic status**	Heterozygous	Heterozygous	Heterozygous
**Ascertainment**	Clinical exome	Research exome	Gene panel
**gnomAD V.2.1.1.**	Absent	Absent	Absent
**Ancestry**	South Asian Indian	European, Ashkenazi Jewish	Chinese
**ClinVar classification as of 2/3/22**	Uncertain significance. Accession: VCV000963526.7	Uncertain significance (4 total: 3 entries (2017–2021) IBMPFD (2 individuals), condition not provided (1 individual), and pathogenic (2018) (1 individual). Accession: VCV000532761.20	Uncertain significance (1 entry IBMPFD). Accession: RCV002301822.1
**MutationTaster (v2021) prediction**	Deleterious Tree vote: 84|16 (del|benign)	Deleterious Tree vote: 87|13 (del|benign)	Deleterious Tree vote: 80|20 (del|benign).
**Conservation between multiple species**	Ile present in 12/12 (total species)	Ala present in 10/12 (total species) Absent in fruit fly, C. elegans	Ile present in 12/12 (total species)
**PolyPhen-2 v2.2.3r406 prediction and score**	Possibly damaging, 1.00	Benign, 0.002	Possibly damaging, 0.873

## Data Availability

All data generated or analysed during this study are included in the published paper.

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
