# Peer review of "Novel Variants in the VCP Gene Causing Multisystem Proteinopathy 1"

_genes, 2023, doi:10.3390/genes14030676_

Round 1

Reviewer 1 Report

Columbres et al. report five patients with novel VCP gene variants. Overall, this is a well-written paper with an excellent discussion of genetic mutations and their potential contribution to MSP1. 

I only have some minor comments:

1. Introduction: The brief representation of FTD is not accurate. That sentence mostly describes the aphasia variants of FTD. The authors have done a great job in describing the FTD spectrum under 1.1 paragraph 3. I propose that they should modify that sentence.

2. "Electromyography (EMG) shows myopathic changes and neuropathic changes including acute and chronic denervation" -> Electromyography shows neuropathy changes including acute and chronic denervation, or myopathic changes, or a combination of both. (or something in that line)

3. Line 81: CT and MRI brain can find cortical atrophy but usually don't show any other structural changes. Postmortem for the diagnosis of FTD is historical, and almost never done anymore. What about PET scan?

4. Line 85: remove cerebrospinal tract. corticospinal tract carries the upper and lower motor neurons. Can be rewritten as "neuronal degeneration of the corticospinal tract affecting both upper and lower motor neurons."

5. line 90: Painless asymmetric limb weakness.

6. Figure 2A. The scapular winging is not well visualized as the patient is wearing an undershirt. If possible please provide an image of the back without the undershirt. 

7. Line 146: spontaneous activity should be mentioned along with muscle activation. Reduced recruitment cannot be predominant. 

Consider rewriting the EMG report as: " EMG revealed spontaneous activities in the form of fibrillation potentials. There was a mix of large and small motor units and recruitment was reduced in most of the muscles examined. However, early recruitment was noted in vastus lateralis (?right/left/both side).

8. Figure 2C. Remove the stars. They are not needed. From the MRI, essentially all the muscles are severely affected. 

9. Line 205: Was the genetic testing a panel of genes? Was whole exome sequencing performed?

10. Line 267: MRI technically cannot show chronic myopathy. Just say that it showed severe muscular atrophy and fatty replacement of ....

11. Line 290: Typo: Generalized fatty muscle atrophy -> generalized muscle atrophy and fatty infiltration

Reviewer 2 Report

My comments are in the manuscript

Reviewer 3 Report

The authors reported the clinical and genetic analysis findings in five patients, 3 from the same family with novel VCPgene variants, associated with cardinal multisystem proteinopathy 1 (MSP1) manifestations including myopathy, PDB, and FTD. They added to the spectrum of heterozygous pathogenic variants found in the VCP gene, and the high degree of clinical heterogeneity. This is an interesting and well-written case report, but there are a few concerns with this paper.

1) With regard to the presented muscle pathology, the authors should mention whether there were any findings suggestive of ALS or neuropathic complications.

2) Skeletal muscle MRI is considered diagnostically useful and should be presented in all cases.

3) The case with VCP (p.Ile369Thr) mutation appears to have a relatively old age of onset, but the mechanism should be discussed
